# Frozen LLM Column Embeddings Are Strong Baselines for Column Type Annotation

**Ehsan Hoseinzade** [1]   **Ke Wang** [1]

## Abstract

Column type annotation (CTA), the task of assigning semantic labels such as Name, City, or Date to table columns, is central to automatic understanding of relational tables. Existing approaches often rely on handcrafted features, task-specific fine-tuning, or prompt-based inference. This paper studies a simpler alternative: using frozen large language models (LLMs) to produce column embeddings and training lightweight classifiers on top. Across six benchmark datasets and ten baselines, frozen LLM column embeddings outperform the strongest prior baseline by 3.7 points without any LLM fine-tuning. Results further show that much of the gain comes from the embedding layer itself, while classifier choice contributes only smaller additional improvements. We further analyze pooling strategies and LLM backbones to better understand the design choices behind frozen LLM-based column embeddings. Overall, strong CTA performance can be achieved without task-specific fine-tuning or prompt engineering, providing a practical baseline for column type annotation.

## 1. Introduction

Column type annotation (CTA) assigns semantic labels such as *Name*, *City*, or *Date* to table columns. CTA is an important component of different information retrieval tasks like data integration, data cleaning, schema matching, and data discovery (Hai et al., 2023; Kandel et al., 2011; Rahm & Bernstein, 2001; Fernandez et al., 2018). One emerging application, for example, is automatically tagging sensitive columns in a table, such as personal information, before deciding what information can be released.

Existing CTA methods mainly fall into three groups. Early deep learning models such as Sherlock and SATO rely on handcrafted features, which are efficient but limited in semantic expressiveness (Hulsebos et al., 2019; Zhang et al., 2019). Later methods improve performance by fine-tuning pretrained language models such as BERT on tabular data (Deng et al., 2020; Suhara et al., 2022; Hoseinzade & Wang, 2024; Sun et al., 2023), but these models remain constrained by limited context length, task-specific fine-tuning cost, and weaker semantic representations than modern large language models (LLMs). More recent LLM-based approaches use prompting or generative fine-tuning (Korini & Bizer, 2023; Kayali et al., 2024; Feuer et al., 2024; Hoseinzade & Wang, 2025; 2026), reducing reliance on feature engineering but often introducing higher cost, prompt sensitivity, and less stable classification behavior.

Meanwhile, recent work in NLP suggests that frozen LLM embeddings paired with lightweight classifiers can be highly effective for classification tasks (Wang et al., 2024; Nie et al., 2024; Su et al., 2023). This raises a simple question for CTA: how far can frozen LLM column embeddings go before more complex task-specific modeling is needed?

This paper studies that question. We represent each table column using embeddings extracted from a frozen LLM and train lightweight classifiers on top for CTA. We compare three classifier families with different structural assumptions: an MLP that predicts each column independently, a conditional random field (CRF) that models local dependencies between adjacent columns, and a graph neural network (GNN) that models dependencies among all columns in a table. Across six benchmark datasets and ten baselines, we find that frozen LLM column embeddings provide a remarkably strong foundation for CTA, consistently outperforming handcrafted-feature pipelines, fine-tuned BERT-based models, and generative LLM approaches. We further analyze pooling strategies, classifier choices, and LLM backbones, and show that mean pooling and mid-sized open-source LLMs already provide most of the benefit.

These results suggest that a large part of the performance gain in modern CTA pipelines can come from the embedding layer itself, making frozen LLM column embeddings a simple, effective, and practical approach that avoids task-specific fine-tuning, prompt engineering, and costly genera-

---

[1]Simon Fraser University, Burnaby, Canada. Correspondence to: Ehsan Hoseinzade <ehoseinz@sfu.ca>.

*Proceedings of the 2nd ICML Workshop on Foundation Models for Structured Data*, Seoul, South Korea. 2026. Copyright 2026 by the author(s).

tive inference.

In summary, this paper makes the following contributions:

- We show that frozen LLM-based column embeddings are a strong baseline for column type annotation, with the best configuration outperforming the strongest baseline by 3.7 micro-F1 points on average across six datasets (Table 2).

- We find that GNN heads perform best (90.2) but the gap across classifier heads is small (88.2–90.2), suggesting the embedding layer drives most of the performance gain (Table 2).

- We provide practical guidance for CTA through an empirical analysis of pooling strategies, and LLM backbones.

## 2. Problem Definition

We study *column type annotation* (CTA), the task of assigning a semantic type to each column of a table based on its cell values. Examples of semantic types include *Name*, *Age*, and *Country*; these differ from atomic data types such as string or integer.

Let $D$ be a collection of labeled tables used for training. Each table $t = (c_1, c_2, \ldots, c_n)$ consists of $n$ columns, where the number of columns and rows may vary across tables. Each column $c_i$ is associated with one label from a predefined set of $k$ semantic types.

A common approach to CTA is to first represent each column $c_i$ with a feature vector (embedding) $\psi_i \in \mathbb{R}^d$, obtained by applying a feature extractor $\phi$ to the values of that column:

$$\psi_i = \phi(c_i).$$

Given the sequence of column embeddings for a table,

$$\psi = \langle \psi_1, \psi_2, \ldots, \psi_n \rangle,$$

the goal is to learn a mapping $f$ that predicts the semantic type of each column:

$$f(\psi) \to \langle y_1, y_2, \ldots, y_n \rangle,$$

where each $y_i$ is one of the $k$ predefined semantic types.

## 3. Approach

We study a simple two-stage pipeline for column type annotation, referred to as **ColEmb**, that combines frozen LLM-based column embeddings with lightweight classifier heads. In the first stage, each column is encoded independently using a frozen LLM. In the second stage, a classifier predicts semantic labels from these embeddings. To study how much structure is needed on top of frozen embeddings, we evaluate three classifier heads with different assumptions about inter-column dependencies: an MLP, a CRF, and a GNN.

### 3.1. Frozen Column Embeddings

Given a table $t = (c_1, \ldots, c_n)$, we represent each column $c_i$ with an embedding $\psi_i = \phi(c_i) \in \mathbb{R}^d$, where $\phi$ is a frozen decoder-only LLM.

To construct the input to $\phi$, we sample $m$ non-null cell values from column $c_i$ and concatenate them into a single text sequence:

$$x_i = \text{concat}(r_1, r_2, \ldots, r_m), \tag{1}$$

where each $r_j$ is a sampled cell value.

The sequence $x_i$ is tokenized and fed into the LLM, producing hidden states from the final transformer layer:

$$\mathbf{H}_i = \text{LLM}(x_i) \in \mathbb{R}^{T_i \times d}, \tag{2}$$

where $T_i$ is the token length and $d$ is the hidden dimension.

We consider two strategies for converting token-level representations into a single column embedding $\psi_i$.

**Mean pooling.** We average the token representations:

$$\psi_i = \frac{1}{T_i} \sum_{l=1}^{T_i} \mathbf{H}_{i,l}. \tag{3}$$

**Last-token pooling.** We use the final token representation:

$$\psi_i = \mathbf{H}_{i,T_i}. \tag{4}$$

Applying $\phi$ to all columns in a table yields the sequence of column embeddings

$$\psi = \langle \psi_1, \ldots, \psi_n \rangle, \tag{5}$$

which serves as input to the classifier head.

### 3.2. Classifier Heads

To isolate the effect of the embedding layer, we evaluate three classifier families on top of the same frozen LLM embeddings.

#### 3.2.1. MLP HEAD

The first classifier treats columns independently. Each embedding $\psi_i$ is passed through a multi-layer perceptron (MLP) that outputs a score vector over the $k$ semantic types:

$$\mathbf{s}_i = \text{MLP}(\psi_i) \in \mathbb{R}^k. \tag{6}$$

Following Sherlock (Hulsebos et al., 2019), we use an MLP with two hidden layers of size 500, Batch Normalization, Dropout, and ReLU activations. The predicted label is

$$\hat{y}_i = \arg\max_j \text{softmax}(\mathbf{s}_i)_j. \qquad (7)$$

### 3.2.2. CRF HEAD

To capture local dependencies between adjacent columns, following SATO (Zhang et al., 2019), we place a linear-chain Conditional Random Field (CRF) on top of the MLP logits. Given logits $\mathbf{s}_i$ for each column $c_i$, the CRF defines the conditional probability of a label sequence $y = (y_1, \ldots, y_n)$ as

$$P(y \mid \psi) = \frac{\exp\left(\sum_{i=1}^{n} \mathbf{s}_i[y_i] + \sum_{i=2}^{n} T_{y_{i-1}, y_i}\right)}{Z(\psi)}, \qquad (8)$$

where $T \in \mathbb{R}^{k \times k}$ is a trainable transition matrix and $Z(\psi)$ is the partition function. Prediction is performed by

$$\hat{y} = \arg\max_y P(y \mid \psi). \qquad (9)$$

### 3.2.3. GNN HEAD

To model table-wide dependencies, following prior graph-based CTA work (Hoseinzade & Wang, 2024; Hoseinzade et al., 2026), we also evaluate a Graph Attention Network (GAT) as the GNN head. The table is represented as a fully connected graph in which each column is a node initialized with its embedding $\psi_i$.

The GAT updates all node representations jointly through attention-based message passing. We use two graph attention layers: the first has 1024 hidden units and the second outputs $k$ logits per node. The GAT produces

$$\mathbf{S} = \langle \mathbf{s}_1, \ldots, \mathbf{s}_n \rangle = \text{GAT}(\psi), \qquad \mathbf{s}_i \in \mathbb{R}^k, \qquad (10)$$

where $\mathbf{s}_i$ is the score vector for column $c_i$. Final predictions are obtained independently from the node-level logits:

$$\hat{y}_i = \arg\max_j \text{softmax}(\mathbf{s}_i)_j. \qquad (11)$$

## 4. Evaluation

### 4.1. Evaluation Method

#### 4.1.1. DATASETS

We use six datasets: T2D (Chen et al., 2019), SOTAB$_{\text{sch}}$, SOTAB$_{\text{sch-s}}$, and SOTAB$_{\text{dbp}}$ (sot, 2023), Wikitable (Deng et al., 2020), Webtables (Sun et al., 2023; Hoseinzade & Wang, 2024; Zhang et al., 2019) summarized in Table 1. We use the same train/test splits and 5-fold cross-validation settings as prior work.

*Table 1.* Summary of datasets. Webtables uses 5-fold cross-validation, and we report the total number of tables.

| Dataset | # Class | # Training Tables | # Test Tables |
|---|---|---|---|
| SOTAB$_{\text{sch}}$ | 82 | 44,769 | 609 |
| SOTAB$_{\text{sch-s}}$ | 82 | 10,631 | 609 |
| SOTAB$_{\text{dbp}}$ | 46 | 37,631 | 279 |
| T2D | 37 | 160 | 109 |
| Wikitable | 255 | 397,098 | 4,764 |
| Webtables | 78 | 78,733 (cross-validation) | |

#### 4.1.2. METRICS AND ALGORITHMS EVALUATED

We evaluate the performance using *micro F1-score* collected on test tables, following previous works (Feuer et al., 2024; Korini & Bizer, 2023; 2024; Hoseinzade et al., 2026; Hoseinzade & Wang, 2026; 2025). All the F1-score values are multiplied by 100 (e.g., 80% is written as 80).

**ColEmb:** We sample 25 rows per column and use the Mistral model (Jiang et al., 2023) with mean pooling to generate column embeddings, which are then classified using three lightweight heads: MLP, CRF, and GNN. Additional results using other LLMs (Mixtral, LLaMA, Qwen, TinyLLaMA) as backbones, and embedding strategies (mean vs. last-token) are provided in Appendix B.2 and B.1, with hyperparameters in Appendix A.

**Baselines:** We compare ColEmb with models from three categories: Deep Learning (DL), Language Models (LM), and Large Language Models (LLM). For each model we used their source code with the mentioned hyper-parameters.

*DL:* Sherlock (Hulsebos et al., 2019) and SATO (Zhang et al., 2019) rely on handcrafted statistical and textual column features. Sherlock uses an MLP classifier, while SATO adds a linear-chain CRF to model dependencies between adjacent columns.

*LM:* TURL (Deng et al., 2020), Doduo (Suhara et al., 2022), RECA (Sun et al., 2023), GAIT (Hoseinzade & Wang, 2024), and REVEAL (Ding et al., 2025) fine-tune BERT-based models for CTA. TURL pre-trains BERT for table understanding; Doduo predicts column types by encoding full tables; RECA incorporates inter-table context; GAIT extends RECA with a GNN over columns; and REVEAL selects columns most relevant to the target column before prediction.

*LLM:* We include representative GPT-based zero/few-shot prompting baselines, GPT$_{\text{ZS}}$ and GPT$_{\text{FS}}$ (Korini & Bizer, 2023), as well as ArcheType (Feuer et al., 2024), which fine-tunes a Llama2-7B for CTA.

### 4.2. Results

**Baseline Comparison.** Table 2 reports micro-F1 scores across six datasets. Traditional deep learning baselines such

*Table 2.* Main results: Micro F1-score comparison of baseline models across six datasets. Models are grouped into three categories: DL (Deep Learning), LM (Language Models), and LLM (Large Language Models). ColEmb outperforms all baselines across all datasets.

| Category | Model | $SOTAB_{sch}$ | $SOTAB_{sch-s}$ | $SOTAB_{dbp}$ | T2D | Wikitable | Webtables | Average |
|---|---|---|---|---|---|---|---|---|
| DL | Sherlock | 80.9 | 72.6 | 77.5 | 67.7 | 75.8 | 87.9 | 77.1 |
| | SATO | 82.1 | 74.2 | 79.0 | 70.7 | 76.0 | 92.5 | 79.1 |
| LM | TURL | 81.7 | 75.4 | 76.7 | 94.0 | 78.9 | 93.1 | 83.3 |
| | RECA | 84.9 | 81.9 | 83.1 | 86.5 | 74.2 | 93.3 | 84.0 |
| | GAIT | 85.8 | 83.1 | 84.7 | 85.1 | 75.5 | 94.1 | 84.7 |
| | Doduo | 86.3 | 81.1 | 85.2 | 91.1 | 75.2 | 96.4 | 85.8 |
| | REVEAL | 86.6 | 83.1 | 86.1 | 91.7 | 75.8 | 95.9 | 86.5 |
| LLM | $GPT_{ZS}$ | 64.0 | 64.0 | 70.0 | 89.4 | 51.7 | 46.3 | 64.2 |
| | $GPT_{FS}$ | 68.2 | 68.2 | 72.4 | 92.1 | 54.5 | 50.1 | 67.6 |
| | ArcheType | 85.1 | 83.0 | 83.6 | 88.0 | 76.7 | 80.3 | 82.8 |
| | **ColEmb$_{MLP}$** | **86.8** | **84.7** | **87.6** | **97.0** | 78.8 | 94.1 | **88.2** |
| | **ColEmb$_{CRF}$** | **87.6** | **85.5** | **89.0** | **97.0** | **79.4** | 94.6 | **88.9** |
| | **ColEmb$_{GNN}$** | **89.2** | **86.2** | **91.5** | 96.2 | **81.0** | 96.8 | **90.2** |

as Sherlock and SATO perform worst on average, reflecting the limitations of handcrafted features. BERT-based CTA models improve substantially, with REVEAL achieving the strongest average performance among LM baselines (86.5). Prompting-based GPT baselines remain much weaker due to lack of supervised training, while ArcheType shows that fine-tuned generative LLMs are stronger than prompting alone but still trail the best discriminative approaches.

**ColEmb vs. Baselines.** All three ColEmb variants outperform the baselines overall, with ColEmb$_{GNN}$ achieving the best average micro-F1 of 90.2. Compared to the strongest LM baseline, REVEAL, this corresponds to a gain of +3.7 points. ColEmb$_{GNN}$ also improves over ArcheType by +7.4 points while requiring no LLM fine-tuning at all. These results show that frozen LLM embeddings, combined with lightweight discriminative classifiers, provide a strong alternative to both BERT fine-tuning and generative label prediction.

**Role of the Embedding Layer.** Across the three ColEmb variants, the performance differences are relatively small: ColEmb$_{MLP}$, ColEmb$_{CRF}$, and ColEmb$_{GNN}$ achieve average micro-F1 scores of 88.2, 88.9, and 90.2, respectively. This confirms that frozen LLM-based column embeddings already provide a strong foundation for CTA, even with simple classifier heads. At the same time, structured heads still offer additional gains: CRF improves over MLP by modeling local dependencies between adjacent columns, while GNN performs best overall by capturing table-wide relationships.

This interpretation is reinforced by matched comparisons with closely related baselines, which further highlight the role of the embedding layer. ColEmb$_{MLP}$ improves over Sherlock by +11.1 points on average (88.2 vs. 77.1),

ColEmb$_{CRF}$ improves over SATO by +9.8 points (88.9 vs. 79.1), and ColEmb$_{GNN}$ improves over GAIT by +5.5 points (90.2 vs. 84.7). These matched comparisons isolate the effect of replacing handcrafted features or intermediate prediction distributions with frozen LLM-based column embeddings, showing that much of the gain comes from the embedding layer itself.

**Practical Defaults.** In addition to the main benchmark, our appendix experiments suggest two practical defaults. First, mean pooling consistently outperforms last-token pooling, improving average micro-F1 by +1.5 points across backbones and classifiers (see appendix B.1). Second, the gain from scaling beyond 7–8B parameters is small: Mistral-7B and LLaMA3-8B perform nearly the same as Mixtral while being much cheaper. These findings suggest that the representational capacity needed for CTA saturates well below the frontier of LLM scale, indicating structured data annotation may be less demanding of model scale than generation tasks (see appendix B.2).

## 5. Conclusion

This work shows that strong CTA performance does not require task-specific LLM fine-tuning or prompt-based inference. Across six benchmark datasets, frozen LLM-based column embeddings paired with lightweight classifiers outperform prior baselines. The results further suggest that much of the gain comes from the embedding layer itself: even simple heads perform strongly, while more structured heads provide additional but smaller improvements. Our analysis also identifies mean pooling as a strong default and shows that mid-sized open-source LLMs already capture most of the benefit. Overall, frozen LLM embeddings provide a simple, effective, and practical approach to CTA.

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

# A. Implementation Details

For the multi-label Wikitable dataset (Deng et al., 2020), we select the first label for each column to ensure compatibility with our single-label setting and several baselines. Column embeddings are generated using a quantized version of the open-sourced Mistral model (Jiang et al., 2023). The MLP, CRF, and GNN classifiers are trained on these embeddings using a batch size of 128. Training parameters are as follows: MLP was trained for 50 epochs with a learning rate of 0.001; CRF for 15 epochs with a learning rate of 0.01; and GNN was trained for 50 epochs with a learning rate of 0.001 and is configured with 2 attention heads. Results are reported using the model from the last epoch on test data.

# B. Analysis

### B.1. ColEmb: Pooling Strategies

Table 3 presents the performance of ColEmb using the Mistral LLM across two embedding strategies (mean and last-token) and three classifiers (MLP, CRF, GNN). Mean pooling consistently outperforms last-token pooling for all classifiers, with improvements of +1.2 for MLP, +1.6 for CRF, +1.7 for GNN, and +1.5 in overall average across models. This confirms that averaging token-level representations across sampled rows yields more robust and semantically informative embeddings, particularly in tabular settings where the final token position is often not meaningful.

*Table 3.* Micro F1-score of ColEmb variants using the Mistral LLM. Models are grouped by embedding strategy (last-token vs. mean). The final column reports the average across all classifiers per embedding type.

| Embedding | Classifier | $SOTAB_{sch}$ | $SOTAB_{sch-s}$ | $SOTAB_{dbp}$ | T2D | Wikitable | Webtables | Avg. | Embedding Avg. |
|---|---|---|---|---|---|---|---|---|---|
| **Last** | MLP | 86.8 | 83.2 | 85.2 | 97.0 | 76.1 | 93.5 | 87.0 | |
| | CRF | 86.7 | 84.0 | 85.3 | 97.0 | 76.6 | 93.9 | 87.3 | 87.6 |
| | GNN | 88.2 | 82.7 | 88.5 | 94.7 | 80.5 | 96.4 | 88.5 | |
| **Mean** | MLP | 86.8 | 84.7 | 87.6 | 97.0 | 78.8 | 94.1 | 88.2 | |
| | CRF | 87.6 | 85.5 | 89.0 | 97.0 | 79.4 | 94.6 | 88.9 | 89.1 |
| | GNN | 89.2 | 86.2 | 91.5 | 96.2 | 81.0 | 96.8 | 90.2 | |

### B.2. ColEmb: LLM Size

Table 4 compares different LLM backbones using *mean pooling*. The main takeaway is that increasing model size beyond 7–8B parameters yields only marginal gains for CTA. TinyLLaMA (1.1B) achieves the weakest average score of 88.1, while Qwen-7B reaches 88.2. Performance improves noticeably when moving to stronger mid-sized models: LLaMA3-8B reaches 88.9 and Mistral-7B reaches 89.1. However, scaling up to Mixtral-47B brings only a very small additional gain, achieving the best average score of 89.2, just +0.1 over Mistral and +0.3 over LLaMA3.

*Table 4.* Average micro-F1 scores across six datasets for different LLM backbones using mean pooling. The final column reports the average across MLP, CRF, and GNN.

| LLM | MLP | CRF | GNN | Avg. |
|---|---|---|---|---|
| TinyLLaMA (1.1B) | 87.3 | 88.2 | 88.9 | 88.1 |
| Qwen (7B) | 87.4 | 88.1 | 89.1 | 88.2 |
| Mistral (7B) | 88.2 | 88.9 | 90.2 | 89.1 |
| LLaMA3 (8B) | 88.1 | 88.9 | 89.9 | 88.9 |
| Mixtral (45B) | 88.5 | 88.9 | 90.3 | 89.2 |

