# OpenReview forum: "Frozen LLM Column Embeddings Are Strong Baselines for Column Type Annotation"
_ICML.cc/2026/Workshop/FMSD — FMSD @ ICML 2026 Poster_

### Official Review · Reviewer_5QHb · 2026-05-20
**Review for ICML 2026 Workshop FMSD Submission159.**

**Rating:** 8
**Confidence:** 5

**Review:**

Summary:
The paper proposes a simple approach for column type annotation. The proposed method uses a frozen large language models (LLMs) to produce column embeddings and training lightweight classifiers on the embedding.

Strengths:
The paper and the proposed method is easy to follow. Moreover, the experiment results show the competitiveness of the proposed method.

Detailed Comments:
It would be interesting to see several more easy machine learning models such as ridge and recent table foundation models. With these efficient learners, it would also be interesting to explore a bit on computational perspectives, possibly giving further strengths of the proposed method. It might also be good to discuss a bit more on extensions as there are more tasks, not just the column type annotation (while making some adjustments in sentences to create some spaces).

Justification of Score:
In general, the paper is easy to follow, and has clear objective in the paper.

---

### Official Review · Reviewer_8JYE · 2026-05-20
**ColEmb: solving column type annotation by shallow classifiers over LLM embeddings**

**Rating:** 5
**Confidence:** 3

**Review:**

The paper discusses ColEmb, a simple approach to perform column type annotation.
Task description: given a table, classify each column into a "type." Each benchmark contains a set of tables, all sharing the same pool of possible labels. Six benchmarks are evaluated in the paper, with between 160 and 400k tables per benchmark, each having between 37 and 255 classes among which to predict.
Method description: The paper proposes to simply use a frozen (autoregressive) LLM, extract the hidden states from the output, pool them, and pass the result through shallow classifiers. Three such classifiers are tested:
1) MLP, working independently on each column (i.e., the problem is reduced to a text classification one)
2) Conditional random fields, estimating transition probabilities between adjacent columns
3) "GNN head" (where however the graph is fully connected, so this appears to be simply a transformer encoder?)

The authors compare against multiple alternative approaches from the literature, issuing from more traditional deep learning with feature engineering, fine-tuned language models (typically BERT-like encoders), or leveraging large language models—the same setting as ColEmb. All three versions of ColEmb score better than all baselines on average, and the order above corresponds in ranking from worst to best.

The main strength of the paper lies in the evaluation scores: I do not have enough experience with the CTA task to be able to judge the provided numbers, but at face value the scores are clearly better than those from multiple baselines, so there must be merit in the idea. However, there are a number of shortcomings that prevent me from giving a good score to the paper.

The main one is that everything in the paper feels outdated in today's AI landscape. The tested LLMs (Mistral, Mixtral, TinyLLaMA, LLaMA3-8B, Qwen) are all very far from the current state of the art. They all came out at least two years ago, which is a very long time in LLM research. Repeating the same experiment with modern open-source models like Gemma 4, Qwen 3, etc., is very likely to lead to significantly better results. The same is true for the baselines: the reported numbers for zero-shot and few-shot usage of a (closed-weights, via API) LLM are for the very first version of ChatGPT (using GPT-3.5), which is certainly far from the current state of the art. Repeating the same analysis, while costly, would give better insights.

Or maybe not—and here comes the second issue that also affects ColEmb: building on pretrained LLMs and evaluating on public data that is widely available (e.g., Wikitable, Webtables) makes it impossible to control for contamination. All of the LLMs used as a basis will most likely have seen this data during pretraining—or maybe it was filtered out because tables are not natural text, but it is impossible to know. Using a sufficiently large modern LLM might even "zero-shot" some classifications correctly if it has memorized them during training.

Another issue I see with the paper (although possibly this goes beyond what is required for a workshop paper) is that the main cited application of CTA and the current methodology also sounds outdated. The authors claim that classifying each column into a "type" can be used to identify sensitive data, e.g., personal information. It is certainly true that a column tagged as "Name" would likely include PII. On the other hand, constructing a dataset with rich labels (such as "Name," "Address," etc.) only to then restrict it to a binary output feels backward. Especially if the solution involves passing data through an LLM, asking directly "please tell me if these sample values appear to contain PII" feels like a much more natural approach than fitting a classifier on many custom labels.

Finally, I have some issues with certain choices and the presentation of the paper:
- Using an autoregressive LLM with average pooling to obtain string embeddings is doable, but dedicated models for string embeddings exist. They are perhaps less popular nowadays, but certainly in the years when the models used in this paper were published, they might have been performing better.
- Except for the MLP head, the other two could be described better. As presented, the CRF approach depends strongly on the specific order of the columns—this needs at least discussion. As mentioned above, the best-performing "GNN head" has really no graph structure, since the graph is fully connected. This would be better presented in terms of a two-layer, unmasked transformer encoder.

Due to the above points, and my limited experience with the CTA task, I have difficulty deciding whether this paper should be accepted for the FMSD workshop. Results are good, but I feel they could be improved easily by building on more modern LLMs (both for the method described here and for at least the two LLM baselines, both dating from 2023). Finally, I am not sure the topic is exactly in scope for the workshop. While it is certainly based on structured data, ultimately the best baselines, as well as at least one of the proposed heads, simply refactor the problem into a text classification one and use standard machinery there. The tabular data is the source of the benchmark data but otherwise plays very little role—there is no real "structure" being used here. As such, my judgment tilts slightly to the negative side, although with low confidence.

---

### Official Review · Reviewer_GTJE · 2026-05-22
**Review of "Frozen LLM Column Embeddings Are Strong Baselines for Column Type Annotation"**

**Rating:** 6
**Confidence:** 3

**Review:**

The experiment reveals a truth: in CTA tasks, the vast majority of the performance leap directly stems from the semantic expressiveness of the LLM embedding layer itself, rather than complex top-level classifier designs, indicating that the performance ceiling of large models in structured annotation tasks is reached relatively early, and 7B-level mid-sized open-source ones capture almost all the gains.